# Efficient Self-Supervised Adaptation of 3D Abdominal Vision-Language Model for Institution-Specific HCC Classification via Full Fine-Tuning and PEFT

**Febryan Putra Kartika**[1]   ID            FBRYNPK@CGU.EDU.TW

**Cheng-Yu Ma**[1,2]   ID                CYMA@CGU.EDU.TW

**Ying-Jia Lin**[1,3]   ID                 YJLIN@CGU.EDU.TW

**Chi-Tung Cheng**[1,4]   ID          ATONG89130@GMAIL.COM

**Kuan-Fu Chen**[1,5]   ID        KFCHEN@GAP.CGU.EDU.TW

[1] *Department of Artificial Intelligence, College of Intelligent Computing, Chang Gung University, Taoyuan, Taiwan*

[2] *Institute of Health Data Science, Chang Gung University, Taoyuan, Taiwan*

[3] *Artificial Intelligence Research Center, Chang Gung University, Taoyuan, Taiwan*

[4] *Department of Trauma and Emergency, Chang Gung Memorial Hospital, Linkou, Taiwan*

[5] *Department of Emergency Medicine, Chang Gung Memorial Hospital, Keelung, Taiwan*

**Editors:** Accepted for publication at MIDL 2026

## Abstract

Medical vision-language models (VLMs) have demonstrated a strong capability in capturing cross-modal relationships between image and text, yet their adaptation to institution-specific clinical tasks remains underexplored. In this study, we fine-tuned a pretrained 3D medical VLM for *hepatocellular carcinoma (HCC) classification* using paired abdominal CT scans and radiology reports from a different institution and with acquisition characteristics that differ from the model's original pretraining corpus. We compared two adaptation strategies: *full fine-tuning* and *parameter-efficient fine-tuning (PEFT)*, motivated by the common use of PEFT to reduce computational cost and enable adaptation under limited-data constraints. Both approaches achieve strong downstream HCC classification performance despite the cross-institutional domain shift, with PEFT reaching an AUC of 0.94 and F1 of 0.91, and full fine-tuning achieving an AUC of 0.95 and F1 of 0.90. These results are competitive with, and in some settings exceed, previously reported supervised HCC classification approaches that rely on lesion-level annotation or segmentation. Full fine-tuning converges rapidly but overfits within a few epochs, whereas PEFT (ConvLoRA for the image encoder and LoRA for the text encoder) attains comparable performance while updating only ∼1% of the model parameters, although requiring more training steps. To better understand adaptation behavior, we also examine the role of contrastive temperature, observing that temperature initialization significantly affects classification performance. This study demonstrates that 3D medical VLM can be efficiently adapted to institution-specific HCC classification using self-supervised CT-report contrastive learning, while highlighting the practical trade-offs between full fine-tuning and parameter-efficient fine-tuning.

**Keywords:** Vision-Language Model, Self-Supervised Learning, Contrastive Learning, Domain Adaptation, Parameter-Efficient Fine-Tuning, Classification

## 1. Introduction

Liver cancer remains a major global health challenge, ranked as the sixth most commonly diagnosed cancer and the third leading cause of cancer-related mortality, with more than 850,000 new cases and more than 750,000 deaths recorded in 2022 (Bray et al., 2024). This burden is dominated by Hepatocellular Carcinoma (HCC), which accounts for approximately 80% of primary liver cancer (Rumgay et al., 2022). New cases and mortality are projected to rise by 31%-98% across different regions by 2045 (Mauro et al., 2025). These trends signal an escalating global health burden, with implications for prevention, early detection, and resource allocation in both high and low-income regions.

A recent systematic review on deep learning methods for HCC (Wei et al., 2023) has shown that conventional deep learning approaches mostly rely on lesion-level annotations or tumor segmentation masks via supervised training, which require substantial radiologist effort and are difficult to scale across institutions. Many existing studies depend on manually outlining the liver lesions across multiple CT phases, making large-scale supervised training costly and limiting the applicability of such models in resource-constrained clinical environments. These requirements motivate the need for approaches that leverage a more efficient form of supervision, such as paired CT scans and radiology reports, which are already available in most routine clinical workflows.

Vision-language models (VLMs) have emerged as a promising paradigm for medical imaging, offering strong generalization without the need for extensive task-specific labels. The CLIP (Radford et al., 2021) framework demonstrated the effectiveness of contrastive image–text alignment, inspiring a wave of radiology-specific VLMs. However, although early applications of vision-language models in the medical domain, such as ConVIRT (Zhang et al., 2022), GLoRIA (Huang et al., 2021) and MedCLIP (Wang et al., 2022) demonstrated the feasibility of aligning visual features with textual descriptions, they were largely driven by 2D frameworks trained on radiographs paired with short reports at sentence-level, despite the inherently 3D and volumetric nature of abdominal CT imaging. In recent years, the field has since progressed toward 3D VLMs capable of processing MRI, CT, and PET volumes. Models like CT-CLIP (Hamamci et al., 2025), HLIP (Zhao et al., 2025), and fVLM (Shui et al., 2025) have demonstrated promising improvements in spatial understanding and clinical relevance. Nevertheless, the computational cost of training and fine-tuning these architectures, coupled with the substantial dataset requirements, remains a major challenge for broad deployment. The recently proposed Merlin (Blankemeier et al., 2024) model addresses several of these limitations by processing complete 3D abdominal CT volumes and leveraging both structured EHR data and unstructured radiology reports for pretraining.

Despite these advances, adapting large medical VLMs to institution-specific downstream tasks remains challenging. Full fine-tuning of 3D VLMs is computationally expensive due to the large size of CT volumes and long radiology reports, and can lead to overfitting, causing the model to fail to adapt to institution-specific data. To address these challenges, we explored parameter-efficient fine-tuning (PEFT) as a lightweight and data-efficient strategy for adapting a pretrained 3D VLM to HCC classification. This choice is motivated by realistic clinical and academic constraints, in which memory, compute, and deployment limitations often preclude extensive retraining of large 3D models where PEFT provides a bounded and stable adaptation strategy that preserves pretrained multimodal representa-

tions while enabling effective domain adaptation. Full fine-tuning is included as a baseline to contextualize the performance and resource trade-offs of PEFT under identical experimental conditions In this work, we applied ConvLoRA (Aleem et al., 2024) on the image encoder and LoRA (Hu et al., 2021) on the text encoder updating only a small fraction of model parameters. We evaluated the effect of temperature initialization, classification performance, and activation heatmaps, examining how PEFT compares to full fine-tuning under limited-data conditions. Although prior medical VLMs adapt CLIP-style (Radford et al., 2021) temperature initialization, the effect of temperature itself has been largely overlooked. Both the 2D radiology VLMs and the recent 3D models mentioned above typically inherit this temperature setting without further analysis. To our knowledge, no existing 3D medical VLM study examines how temperature affects adaptation behavior on downstream classification performance, even though temperature governs how strongly the contrastive loss penalizes hard negative samples by scaling their contribution to the contrastive gradients (Wang and Liu, 2021). Our contributions include:

1. We demonstrated that competitive HCC classification performance can be achieved without lesion segmentation labels or extensive computational resources, using self-supervised contrastive learning and PEFT of a 3D VLM on institutional CT-report pairs

2. We evaluated the performance of PEFT-based adaptation against full fine-tuning, quantifying the trade-offs between predictive accuracy, training stability, and efficiency in HCC classification task.

3. We present the first analysis of contrastive temperature in 3D medical VLM adaptation, showing how temperature reshapes embedding structure, class separability, and downstream HCC classification performance.

4. We release all our trained models and code, enabling full reproducibility and supporting further research on efficient 3D VLM adaptation.

This work highlights a practical and computationally efficient path closer toward hospital-specific deployment of medical foundation models under real-world data constraints. Code available at: https://github.com/fbrynpk/HCC-Merlin

## 2. Materials and Methods

### 2.1. Dataset

This study uses an internal, institution-specific abdominal CT dataset containing contrast-enhanced multiphase abdominal CT scans and corresponding radiology reports. The full dataset comprises four contrast phases with the following distribution: non-contrast (24.6%), arterial phase (25.5%), portal venous (26.6%), and delayed phase (23.3%). For all experiments, we restrict our analysis to the **portal venous (PV) phase**, as Merlin (Blankemeier et al., 2024) was pretrained predominantly on PV-phase abdominal CT scans, making PV inputs most aligned with its pretraining distribution and reducing cross-phase domain shift allowing a more controlled evaluation to better assess adaptation behavior. Additionally,

preliminary experiments incorporating multi-phase inputs did not yield consistent performance improvements in our setting. As such, and to avoid introducing additional confounding factors, we restrict the present study to PV-phase imaging. Comprehensive multi-phase modeling is left for future work. Furthermore, PV-phase scans are also the most consistently available in our institutional cohort.

After filtering the full dataset to PV-phase studies, the resulting cohort contains 3,611 scans with 1,713 HCC-positive and 1,898 negative cases. The negative group consists of patients without hepatocellular carcinoma and does not represent a healthy control population. Negative cases may exhibit chronic liver disease (e.g., fatty liver or chronic hepatitis), benign hepatic findings such as cysts, or other non-malignant abdominal findings commonly encountered in routine clinical imaging. No primary liver tumors other than HCC are included in the dataset. This heterogeneous composition reflects realistic clinical screening conditions and reduces the likelihood of trivial separation between positive and negative cases. **Importantly, HCC labels originated from the dataset and are *not* extracted from the radiology reports. These labels are solely used on the validation and test sets for metrics calculation**.

Table 1: Portal Venous Phase Dataset Distribution (HCC vs Negative)

| Split | Patients | Scans | Slices |
|---|---|---|---|
| Train | 1,281 | 2,129 | 153,753 |
| Validation | 440 | 801 | 54,231 |
| Test | 435 | 681 | 51,235 |
| Total | 2,156 | 3,611 | 259,219 |

**Dataset Splits** We split the dataset at the **patient level** into training, validation and test sets using a 60% / 20% / 20% allocation. Patient-level splitting ensures that no individual appears in more than one subset, preventing data leakage from patients with multiple visits or repeated scans. Because patients contributed varying numbers of studies, phases, and time points, the resulting *sample*-level proportions deviated slightly from the target 60/20/20 split. Phases and CT scans distribution are provided in Appendix A

### 2.2. Dataset Preprocessing

**Image Preprocessing:** All CT volumes were preprocessed following the original Merlin (Blankemeier et al., 2024) pipeline. Each scan was reoriented to RAS+ convention, resampled the in-plane axial images to 1.5 mm resolution, and out-of-plane slice thickness to 3.0 mm spacing using bilinear interpolation. We then mapped the Hounsfield unit (HU) range -1000:1000 to the range 0:1, clipping values that fall outside of this range. Finally, we pad and center-crop to 224 x 224 pixels in-plane and 160 pixels out-of-plane

**Report Preprocessing:** Radiology reports in our dataset are unstructured free text. To obtain consistent, structured input for the text encoder, we employed Qwen3-8B (Team,

2025) to segment each report into **Clinical Information**, **Findings**, and **Impression** sections. We then further decomposed the **Findings**, and **Impression** sections into anatomical subregions using the same prompting strategy based on Merlin's (Blankemeier et al., 2024) original pretraining. This ensures that the textual granularity and anatomical alignment remain consistent with the model's pretraining regime. The exact prompt used for extraction and segmentation is provided in Appendix B.

### 2.3. Base Model (Merlin)

Merlin (Blankemeier et al., 2024) is a 3D vision-language model pretrained specifically for abdominal imaging. The model leverages paired abdominal CT volumes, structured electronic health record (EHR) data, and unstructured radiology reports to learn joint visual-text representations. Its pretraining dataset consists of over 6 million CT slices derived from 15,331 abdominal CT studies, alongside more than 1.8 million EHR codes and over 6 million tokens of radiology report text. For the image encoder, it employs an inflated ResNet-152 (Carreira and Zisserman, 2018), unlike most 3D VLMs that rely on Vision Transformers (Dosovitskiy et al., 2021), which typically demand substantial memory and multi-GPU environments. This convolutional backbone offers a more computationally efficient alternative. This design choice enables full 3D volumetric training on a comparatively modest hardware using only a single A6000 GPU, significantly reducing the resource burden associated with 3D VLMs pretraining. For the text encoder, it uses a Clinical Longformer (Li et al., 2022) selected for its ability to process long radiology reports that exceed the context length of standard transformer architectures.

### 2.4. Adaptation Setup

#### 2.4.1. FULL FINE-TUNING:

In this setting, all parameters of both the image encoder and text encoder are updated. During fine-tuning, we alternate between the **Findings** and **Impression** sections with the anatomical decomposed section per steps following Merlin (Blankemeier et al., 2024) pretraining strategy. Full fine-tuning enables the model to fully adapt to the target HCC classification task but requires significantly more computational resources and carries a higher risk of overfitting with a limited dataset. This baseline reflects the conventional approach to domain adaptation but is computationally expensive and less suitable under limited-data conditions.

#### 2.4.2. PARAMETER-EFFICIENT FINE-TUNING (PEFT): CONVLORA AND LORA:

To reduce the computational and data demands, we apply parameter-efficient fine-tuning (PEFT) by introducing lightweight adapter modules into the pretrained model while keeping the backbone frozen, except for the final projection layers of the text and image encoders, as we found that freezing these layers prevented the VLM from adapting to the new task effectively.

**LoRA - Text Encoder** We first adapt the transformer-based text encoder using LoRA (Hu et al., 2021), applied to the query, key, and value projection matrices within each multi-head self-attention layer. Given a pretrained weight matrix $W_0 \in \mathbb{R}^{d_{\text{out}} \times d_{\text{in}}}$, this introduces

a low-rank residual update:

$$W' = W_0 + \Delta W, \qquad \Delta W = \frac{\alpha}{r} BA$$

where $A \in \mathbb{R}^{r \times d_{\text{in}}}$ and $B \in \mathbb{R}^{d_{\text{out}} \times r}$ are low-rank factors with $r \ll \min(d_{\text{in}}, d_{\text{out}})$, and $\alpha$ is a scaling factor. The pretrained weights $W_0$ remain frozen, and optimization occurs solely through $A$ and $B$. This preserves the representational geometry of the pretrained transformer while allowing efficient adaptation with small parameter updates.

**ConvLoRA - Image Encoder** We then apply ConvLoRA (Aleem et al., 2024) to 3D convolutional layers following the original formulation. Following LoRA, this method decomposes the update to a convolutional kernel into a low-rank residual, enabling task-specific adaptation with a minimal number of trainable parameters. For a given 3D convolutional layer with pretrained weights $W$, ConvLoRA constrains the update by parameterizing it as:

$$W' = W_0 + \Delta W, \qquad \Delta W = BA$$

where $B \in \mathbb{R}^{m \times r}$ and $A \in \mathbb{R}^{r \times n}$ are low-rank matrices with rank $r \ll \min(m, n)$. Similar to LoRA, the original kernel $W_0$ is frozen, and only the low-rank factors $B$ and $A$ are learned during fine-tuning, ensuring that the update lies within a restricted low-dimensional subspace while preserving the pretrained convolutional structure. Following the original work, $A$ is initialized with random Gaussian weights and $B$ is initialized to zero, ensuring that adaptation begins from the pretrained representation. For more details on $r$ & $\alpha$, refer to Appendix C

Table 2: LoRA & ConvLoRA Configuration

| Component | LoRA | ConVLoRA |
|---|---|---|
| Injection layers | $Q$, $K$, $V$ projections in all [**12**] layers | [**Layer1, Layer2, Layer3, Layer4**] |
| Rank ($r$) | **16** | **2** |
| Scaling factor ($\alpha$) | **32** | **2** |
| Trainable params | 884,736 / 149,274,368 | 916,992 / 121,882,204 |

### 2.4.3. Loss Function & Training Hyperparameters

We adopt a CLIP-style (Radford et al., 2021) self-supervised contrastive learning based on the InfoNCE loss (van den Oord et al., 2019) using CT scans and medical reports to align the image and text embeddings produced by the image and text encoders. Given a batch of 18 image-text pairs, the image features $\{v_i\}_{i=1}^N$ and text features $\{t_i\}_{i=1}^N$ are projected into a shared embedding space and normalized. Similarity is computed using the cosine similarity scaled by a learnable temperature $\tau$. The symmetric InfoNCE loss is defined as:

$$\mathcal{L}_{\text{InfoNCE}} = -\frac{1}{2N} \sum_{i=1}^{N} \left[ \log \frac{\exp(\text{sim}(v_i, t_i)/\tau)}{\sum_{j=1}^{N} \exp(\text{sim}(v_i, t_j)/\tau)} + \log \frac{\exp(\text{sim}(t_i, v_i)/\tau)}{\sum_{j=1}^{N} \exp(\text{sim}(t_i, v_j)/\tau)} \right],$$

where $\text{sim}(\cdot, \cdot)$ denotes cosine similarity. This loss encourages matched image-text pairs to remain close while pushing apart mismatched pairs. Full fine-tuning and PEFT were trained using identical optimization settings to enable direct comparison. Details on the hyperparameter settings are listed in Appendix C

### 2.4.4. ZERO-SHOT CLASSIFICATION

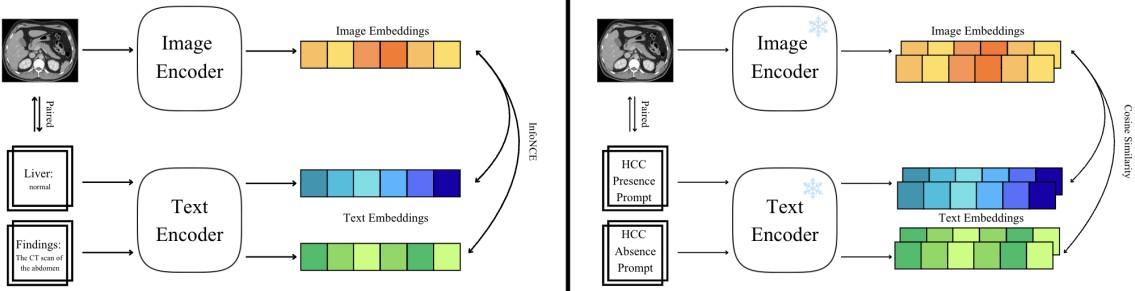

Figure 1: *Left:* Overview of Self-Supervised Fine-Tuning *Right:* Overview of Zero-Shot Classification

To perform zero-shot classification in Figure 1, we follow the standard vision-language matching paradigm used in contrastive VLMs. The CT volume is encoded into a 3D image embedding, while each textual prompt representing either HCC or a negative description is encoded into the text embedding. Classification is then performed by computing the cosine similarity between the image embedding and each text embedding. The model assigns the class whose prompts yield the highest similarity score. Because cosine similarity directly reflects alignment in the joint embedding space, the model effectively selects the class description it believes best matches the input CT volume. The final prediction is obtained by aggregating similarities across multiple prompts per class.

$$HCCCosineSimilarity < NegativeCosineSimilarity = NegativePrediction$$

## 3. Results

### 3.1. Zero-shot baseline

A recent systematic review of deep learning methods for HCC classification reported an average sensitivity of 0.89 (95% CI [0.87-0.91]), specificity of 0.90 (95% CI [0.93-0.97]), and AUC of 0.95 (95% CI [0.93-0.97]) across supervised approaches requiring lesion annotations or curated training labels (Wei et al., 2023). Due to the lack of publicly available supervised deep learning models and datasets, our evaluation begins with zero-shot classification without any liver-specific supervision using pretrained vision-language models. Following Merlin's (Blankemeier et al., 2024) comparison, we used BioMedCLIP (Zhang et al., 2025), a VLM trained on 15 million 2D biomedical images-text pairs from scientific articles, which performs poorly in this 3D volumetric CT setting with an F1 of 0.143 (95% CI [0.090-0.191]).

The pretrained Merlin checkpoint achieves a substantially stronger baseline with an F1 of 0.607 (95% CI [0.565-0.648]), yet still underperforms compared to supervised HCC literature benchmarks. These results highlight the intrinsic difficulty of liver tumor detection in 3D CT without targeted adaptation and motivate the need for domain-specific tuning.

Table 3: Result Comparison in Testing Set

| Model | F1↑ | Recall↑ | Precision↑ | Accuracy↑ | AUC↑ |
|---|---|---|---|---|---|
| BioMedCLIP | 0.14 | 0.10 | 0.26 | 0.49 | 0.16 |
| Merlin (Base) | 0.61 | 0.70 | 0.54 | 0.61 | 0.67 |
| Merlin (PEFT) | 0.91 | 0.95 | 0.88 | 0.92 | 0.94 |
| Merlin (Full Fine-Tuned) | 0.90 | 0.96 | 0.85 | 0.91 | 0.94 |

## 3.2. Fine-tuning vs PEFT

Table 4: Comparison of PEFT and Full Fine-Tuning across Different Contrastive Temperatures ($\tau$). Metrics reported: F1, Recall, Precision, Accuracy, AUC, Mean Cosine Similarity between image-text pairs.

| Temp. Init. ($\tau$) | Method | F1↑ | Recall↑ | Precision↑ | Accuracy↑ | AUC↑ | CosSim↑ |
|---|---|---|---|---|---|---|---|
| 0.07 | PEFT | 0.84 | 0.85 | 0.84 | 0.86 | 0.92 | 0.46 |
| 0.07 | Full FT | 0.81 | 0.85 | 0.77 | 0.83 | 0.90 | 0.26 |
| 0.5 | PEFT | 0.87 | 0.95 | 0.83 | 0.90 | 0.95 | 0.64 |
| 0.5 | Full FT | 0.88 | 0.96 | 0.82 | 0.89 | 0.94 | 0.60 |
| 1.0 | **PEFT** | **0.91** | **0.95** | **0.88** | **0.92** | **0.94** | 0.77 |
| 1.0 | Full FT | 0.89 | 0.97 | 0.83 | 0.90 | 0.95 | 0.68 |
| 10.0 | PEFT | 0.91 | 0.94 | 0.88 | 0.92 | 0.93 | 0.83 |
| 10.0 | **Full FT** | **0.90** | **0.96** | **0.85** | **0.91** | **0.95** | 0.80 |

Both adaptation strategies substantially improve HCC classification over the zero-shot baseline. However, PEFT provides slightly better downstream performance and more stable training than Full Fine-tuning (refer to Appendix D). Our best settings yield a zero-shot F1 of 0.912 (95% CI [0.887-0.935]) for PEFT with an initial $\tau$ of 1.0 and an F1 of 0.903 (95% CI [0.877-0.926]) with an initial $\tau$ of 10.0 for full fine-tuning. Our performance is comparable to multiple deep learning methods reviewed in 2023 (Wei et al., 2023) that needed label supervision or lesion segmentation for training. An ablation study was conducted for the PEFT methods in Appendix E.

### 3.3. Effects of Contrastive Temperature

We further examined the effect of the contrastive temperature ($\tau$) used during adaptation. Temperature directly controls the sharpness of the similarity distribution in the contrastive softmax (Figure 12). Lower initial values produce sharper peaks, while higher initial values yield softer and more diffuse distributions. Empirically, from Figure 2 we observe that at the lowest initial temperature $\tau = 0.07$, the embeddings exhibit less stable clustering, with noticeable overlap between HCC and non-HCC samples. As the initial temperature increases, the embedding space becomes progressively more organized, samples form smoother and more coherent clusters, hence the separation between positive and negative classes becomes more pronounced. The highest initial temperature $\tau = 10.0$ produces the most visible inter-class separation, reflecting improved embedding uniformity and reduced overfitting. These embeddings align with the training-validation loss behavior (Figure 6), indicating that higher initial temperatures promote more stable representation learning and better class separability. This behavior is further illustrated by the cosine similarity matrices in Appendix F.2 and how contrastive temperature affects embedding shifts in Appendix F.3.

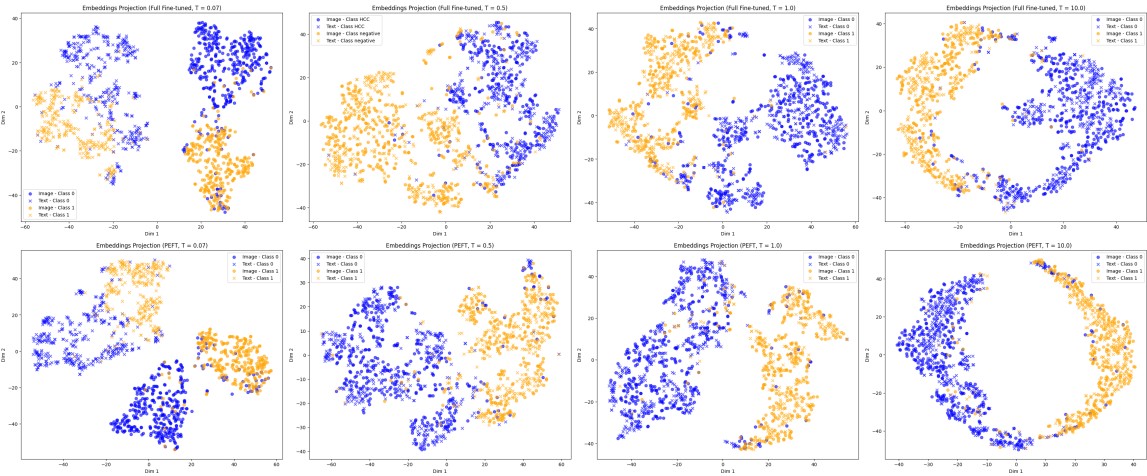

Figure 2: *Upper:* Full Fine-tuned, *Lower:* PEFT Embeddings Shift of Each Temperature Initialization [**0.07, 0.5, 1.0, 10.0**]

### 3.4. Interpretability

To understand how adaptation alters the model's decision process, we applied GradCAM (Gildenblat and contributors, 2021) between a random CT scan and its paired report to visualize the spatial regions most influential to the HCC classification output. Figure 3 shows that in the zero-shot setting, the pretrained model exhibits diffuse attention that often fails to localize to the liver. After adaptation, both full fine-tuning and PEFT redirect the model's focus toward anatomically relevant liver regions, with PEFT producing the strongest and most spatially concentrated responses. This shift indicates that adaptation

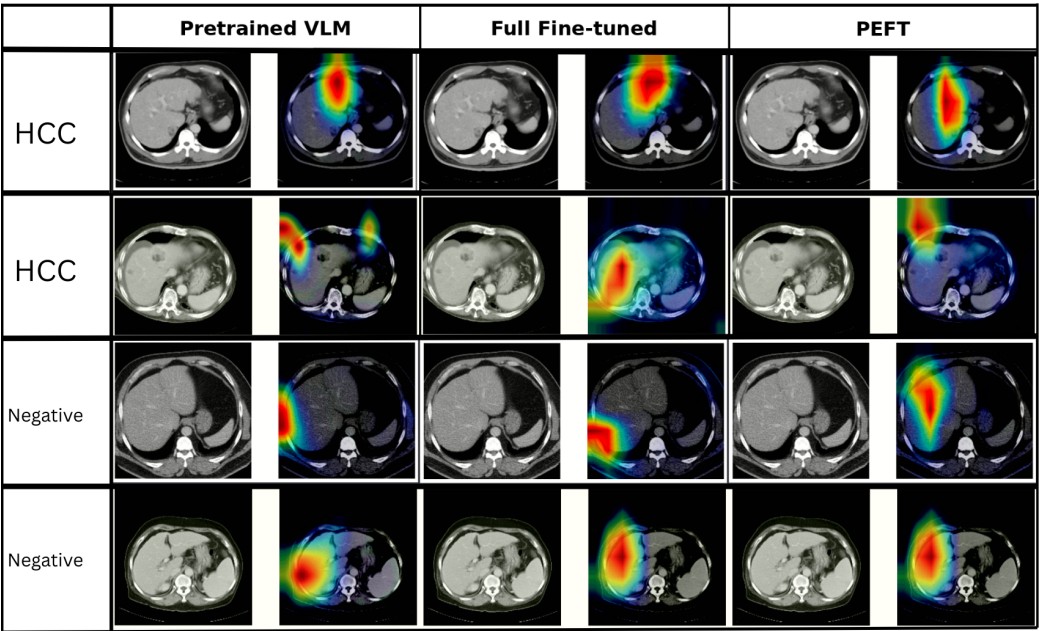

Figure 3: GradCAM heatmaps before and after adaptation.

not only improves classification accuracy but also refines the model's spatial reasoning, suppressing irrelevant contextual cues and emphasizing clinically meaningful features.

### 3.5. External Evaluation on VerSe Spine Fracture Dataset

To assess whether the adaptation methods under limited supervision leads to task-specific overfitting, we additionally evaluated the adapted models on the external VerSe (Löffler et al., 2020) fracture dataset. VerSe consists of CT scans labeled for vertebral fractures and represents a different anatomical region and pathology compared to the target HCC task. Importantly, this dataset was not used for adaptation or hyperparameter tuning in this study. We evaluate the pretrained Merlin model, as well as models adapted via full fine-tuning and PEFT, using the same evaluation protocol. This analysis is intended as a supporting sanity check demonstrating that the proposed adaptation strategy does not degrade behavior on an external task under limited data and compute settings, rather than as a claim of broad task generalization. The adapted model has shown comparable or improved performance on some setting relative to the pretrained baseline, suggesting that the learned representation did not collapse to trivial task-specific cues.

Table 5: External evaluation on the VerSe spine fracture dataset across Different Contrastive Temperatures ($\tau$). Metrics reported: F1, Recall, Precision, Accuracy.

| Temp. Init. ($\tau$) | Method | F1↑ | Recall↑ | Precision↑ | Accuracy↑ |
|---|---|---|---|---|---|
| Baseline | Baseline | 0.77 | 0.88 | 0.68 | 0.65 |
| 0.07 | PEFT | 0.65 | 0.62 | 0.70 | 0.58 |
| 0.07 | Full FT | 0.75 | 0.92 | 0.63 | 0.60 |
| 0.5 | PEFT | 0.75 | 0.85 | 0.67 | 0.63 |
| 0.5 | Full FT | 0.76 | 0.92 | 0.65 | 0.63 |
| 1.0 | **PEFT** | **0.76** | **0.85** | **0.69** | **0.65** |
| 1.0 | Full FT | 0.75 | 0.92 | 0.63 | 0.60 |
| 10.0 | **PEFT** | **0.76** | **0.85** | **0.69** | **0.65** |
| 10.0 | **Full FT** | **0.80** | **0.85** | **0.76** | **0.73** |

## 4. Discussion

This study demonstrates that a pretrained 3D medical vision-language model can be effectively adapted for *hepatocellular carcinoma* classification using only paired CT-report data and without the need for pixel-level annotations. Both full fine-tuning and parameter-efficient fine-tuning substantially improved the zero-shot performance of Merlin (Blankemeier et al., 2024), but PEFT achieved the strongest overall results while offering clear practical advantages. By updating only low-rank adapter modules in the image and text encoders, GPU memory usage during training decreased from approximately 48 GB to 20 GB on a single NVIDIA A6000 GPU. This makes training feasible on modest hardware and enables larger effective batch sizes, highlighting that large 3D VLMs can be adapted efficiently even in resource-constrained environments. These results align with the core motivation of this work, enabling practical, institutional-level fine-tuning without requiring large compute clusters, extensive annotation pipelines, and a large curated dataset. Our analysis of contrastive temperature further highlights its important role in downstream performance. For class-level classification tasks, where broad semantic separation is more important than fine-grained alignment, higher temperature initialization produced more stable optimization and stronger overall performance. To our knowledge, this behavior has not been characterized in 3D medical VLM adaptation, underscoring temperature initialization as an underappreciated but critical hyperparameter for stable and effective domain transfer. Interpretability assessment using GradCAM (Gildenblat and contributors, 2021) showed that both adaptation strategies shifted Merlin's attention toward anatomically relevant regions of the liver after training. This demonstrates that paired CT-report self-supervision is sufficient to steer spatial focus toward clinically meaningful structures even without segmentation labels. Despite these promising findings, several limitations should be acknowledged. Our study only evaluates one PEFT strategy (ConvLoRA (Aleem et al., 2024) and LoRA (Hu et al., 2021)), though alternative adapters, prefix tuning, or hybrid

update strategies may offer different trade-offs between accuracy and efficiency. Furthermore, our experiments focus exclusively on portal venous phase CT, whereas multi-phase abdominal imaging could provide complementary diagnostic information and may further improve performance. Finally, we address binary HCC classification, extending this framework to multi-class liver tumor classification would more closely reflect real-world diagnostic workflows and better evaluate the representational capacity of medical VLM adaptation.

## 5. Conclusion

We demonstrated a resource-efficient approach for adapting a pretrained 3D vision-language model to *hepatocellular carcinoma* classification using only paired CT-report data. Parameter-efficient fine-tuning achieved the best overall performance while reducing GPU memory requirements by more than half compared to full fine-tuning, demonstrating that large-scale VLM adaptation is feasible on a single GPU. Our analysis also shows that contrastive temperature substantially influences classification performance, with higher temperature producing more effective class-level separation. GradCAM visualization further confirms that adaptation shifts the model's focus toward relevant liver regions, supporting the plausibility of the learned representations. These results highlight the promise of VLM-based adaptation as a practical alternative to traditional supervised pipelines, enabling strong performance without lesion segmentation or large labeled datasets. This work is not designed to demonstrate cross-institutional generalization. We aim to evaluate whether VLMs can be adapted efficiently in a single-institution scenario using limited hardware, consistent with real-world hospital constraints. Future work will investigate extensions to multi-phase CT and multi-class liver tumor classification.

## Acknowledgments

This work was supported by the National Science and Technology Council in Taiwan (114-2314-B-182-013, 114-2321-B-182-001) and Chang Gung Memorial Hospital in Taiwan (CM-RPG2P0181, CMRPG2P0342, CMRPG2P0121).

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

## Appendix A. CT Phases & Scan Distributions

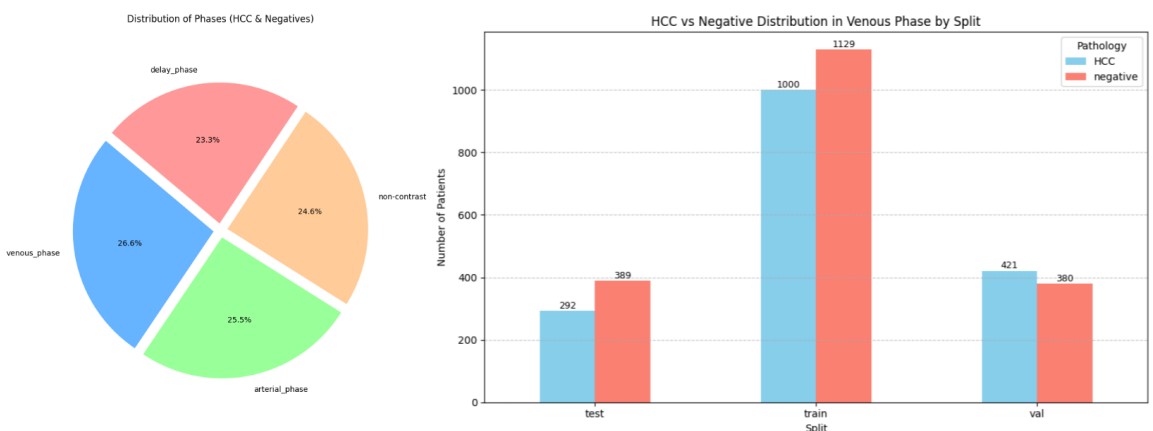

Figure 4: *Left*: CT Phases Distribution. *Right*: HCC vs Negative Distribution

## Appendix B. Text Preprocessing Prompts

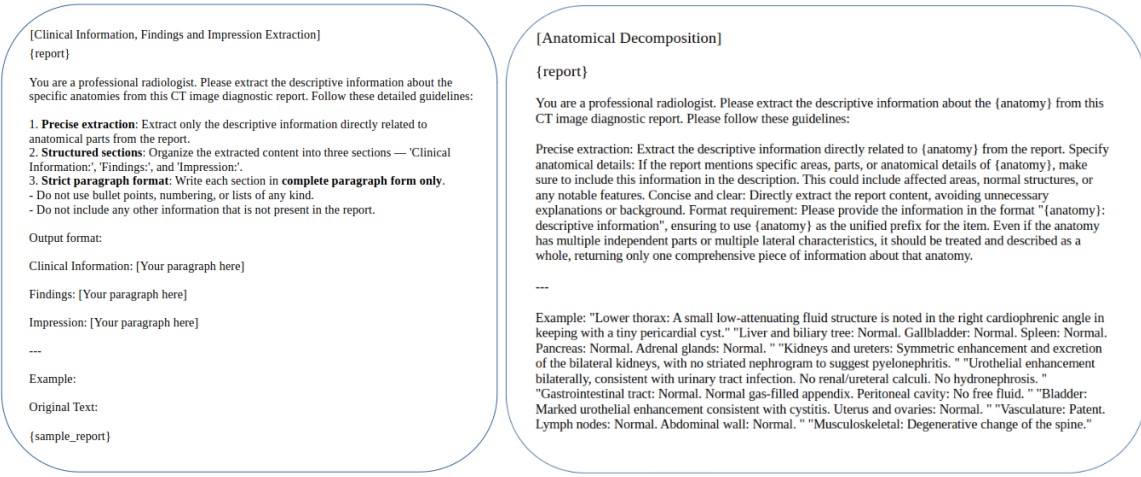

Figure 5: *Left*: Clinical Information, Findings, and Impression Extraction Prompt. *Right*: Anatomical Decomposition Prompt

Figure 5. Illustrates the two prompt designs used for preprocessing: a clinical report extraction prompt that retrieves key sections [**Clinical Information, Findings and Impression**], and an anatomical-decomposition prompt that restructures the report into organ-level descriptions. Both prompts were implemented using a one-shot extraction

strategy, where a single example output follows the instruction. We observed that one-shot prompting consistently produced more structured and reliable outputs than zero-shot prompting.

We also compared Qwen3-8B and Qwen3-14B (Team, 2025) for the extraction process. Despite the larger model size, Qwen3-14B produced outputs with greater variability in formatting and anatomical decomposition, whereas Qwen3-8B generated more stable and consistent extractions across all cases. For this reason, Qwen3-8B was adopted as the primary extractor. To assess whether the extracted text was suitable as supervision for contrastive fine-tuning, we evaluated the outputs using **BERTscore** (Zhang* et al., 2020) and the **GREEN** score (Ostmeier et al., 2024). The extraction achieved a BERTScore of 0.89 and 0.84 for Extraction and Decomposition, respectively and a GREEN score of 0.79 and 0.58, respectively, indicating that the generated narratives preserve the core semantic content and clinical fidelity of the original reports. These results demonstrate that the combination of one-shot prompting and Qwen3-8B provides sufficiently accurate and structured text for reliable supervision during VLM adaptation.

## Appendix C. Training Hyperparameter

Table 6: Training Hyperparameter

| Hyperparameter | Full Fine-Tuning | PEFT (ConvLoRA + LoRA) |
|---|---|---|
| Batch Size | 18 | 18 |
| Gradient Accumulation | 8 | 8 |
| Learning Rate | 1e-5 | 1e-4 |
| Optimizer | AdamW | AdamW |
| Epoch | 300 | 300 |
| Early Stopping Patience | 5 | 5 |
| Trainable Parameters | 271M | 3.4M |
| Temperature ($\tau$) | [0.07, 0.5, 1.0, 10.0] | [0.07, 0.5, 1.0, 10.0] |

The learning rates were selected to match the optimization behavior expected for each adaptation strategy. For full fine-tuning, we use 1e-5, following the learning rate used during Merlin's (Blankemeier et al., 2024) original VLM pretraining and necessary for stable optimization when updating all encoder parameters. Empirically, increasing the learning rate to 1e-4 caused the model to rapidly overfit and diverge from the pretrained representation before reaching optimal performance, so 1e-5 was retained as the only rate that produced stable convergence.

For PEFT, we evaluated both 1e-5 and 1e-4. While the lower rate achieved a similar final performance, it converged noticeably slower and offered no measurable benefit. Because only the adapter parameters and final projection layers are trainable and can tolerate faster updates, we selected 1e-4 as the more efficient and effective learning rate for PEFT.

We set the gradient accumulation step to 8 for both methods to achieve an effectively larger batch size, which is important for contrastive learning. Larger batch sizes provide

more negative samples per update, yielding more stable similarity estimates and more reliable contrastive gradients. Fine-tuning without gradient accumulation resulted in noticeably less stable training dynamics. Accumulating gradients over 8 steps allowed us to maintain stable optimization while preserving memory usage.

Other optimization settings followed the original VLM pretraining setup. We use the **AdamW** optimizer with $\beta = (0.9, 0.999)$. A **cosine learning rate scheduler** is applied with decay to zero over **300 epochs**. Early stopping with a patience set to 5 epochs based on validation F1 score is used to prevent overfitting. To reduce memory usage, **gradient checkpointing** for both the image and text encoders and training in **FP16 mixed precision** were also applied.

For LoRA and ConvLoRA settings in Table 2, we set LoRA $r = 16$ with scaling factor $(\alpha) = 32$, following common practice in VLM and transformer-based adapter literature where moderate ranks provide a strong balance between learning capacity and parameter efficiency. Preliminary experiments with lower ranks (e.g., $r = 4$ or 8) eventually reached comparable performance, though with more training time and exhibited slower convergence, offering no practical benefit under our compute constraints, while higher ranks (e.g,. $r = [32, 64]$) increased memory usage without noticeable performance gains. For ConvLoRA, we adopt both $r$ & $\alpha = 2$ as proposed in the original ConvLoRA formulation. We empirically evaluated higher ConvLoRA ranks ($r = 4, 8, 16$) and observed no consistent performance improvements over $r = 2$, while incurring additional memory overhead. These settings therefore represent an effective trade-off between stability, adaptation capacity, and parameter efficiency.

## Appendix D. Full Fine-Tuning vs PEFT Training

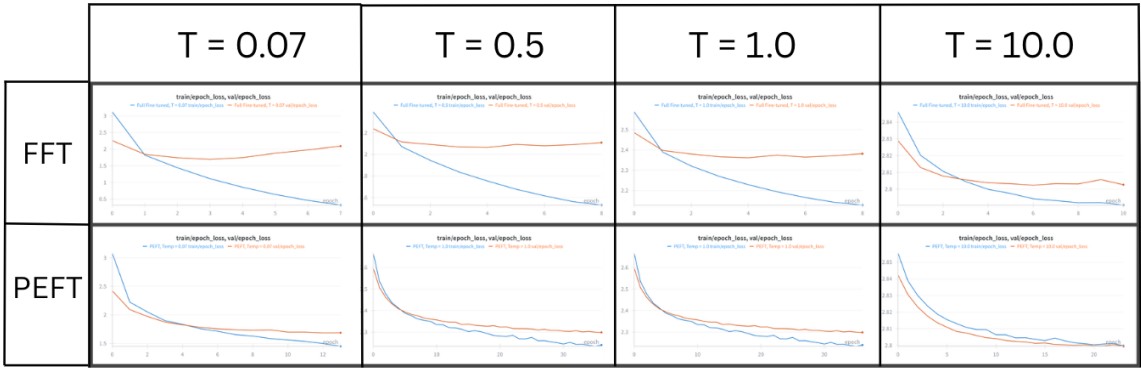

Figure 6: Training-Validation Loss for each setup and temperature *Blue: Training, Orange: Validation*

The training-validation loss curves at Figure 6. highlight how contrastive temperature affects optimization stability. At a low temperature of $\tau = 0.07$, the model updates rapidly due to sharper similarity distributions and larger contrastive gradients. This accelerates

early learning but also leads to rapid overfitting, reflected by the widening train-validation gap. As the temperature increases (e.g,. $\tau = 0.5$ and $\tau = 1.0$), the gradients become smoother, slowing the update rate and reducing the tendency to memorize the training data, although the overfitting is still noticeably visible. At the highest temperature, $\tau = 10.0$, training progresses more gradually and exhibits the smallest divergence between training-validation loss, indicating improved robustness. Across all temperatures, PEFT consistently shows reduced overfitting compared to full fine-tuning, most prominently at $\tau = 0.07$. Because PEFT updates only a small set of low-rank adapter parameters, its effective capacity is lower, which inherently regularizes the optimization process. This results in more stable training dynamics.

## Appendix E. Ablation Study

All ablation experiments from Table 7. were performed on the **validation set** with the best PEFT settings highlighted from Figure 4, with reported metrics corresponding to the best F1 score obtained within 10 epochs of training. We intentionally restrict each ablation run to 10 epochs to capture the *early-phase adaptation behaviour* of each PEFT component. In our full PEFT setting, performance increases rapidly in the first few epochs and then plateaus, with only marginal gain afterward. Each ablation isolates a single adaptation pathway:

- Text Encoder - LoRA applied only to the text encoder and unfrozen final text projection layer. Image encoder entirely frozen

- Image Encoder - ConvLoRA applied only to the image encoder and unfrozen final image projection layer. Text encoder entirely frozen

- Projection Layers - Only the final image and text projection layers are unfrozen. The rest of the backbone remains frozen.

- Full PEFT - All PEFT components active. Represents the complete adaptation strategy and achieves the strongest performance.

Table 7: Ablation Study on PEFT Methods

| Method | F1↑ | Recall↑ | Precision↑ | Accuracy↑ | Parameter (Trainable)↓ |
|---|---|---|---|---|---|
| Baseline | 0.67 | 0.69 | 0.67 | 0.66 | 271M (0) |
| Text Encoder | 0.78 | 0.77 | 0.79 | 0.77 | 271M (1.28M) |
| Image Encoder | 0.88 | 0.93 | 0.84 | 0.87 | 271M (1.96M) |
| Projection Layers | 0.89 | 0.93 | 0.86 | 0.88 | 271M (1.44M) |
| Full PEFT | 0.92 | 0.96 | 0.88 | 0.91 | 271M (3.24M) |

This ablation design determines whether each component contributes unique signal or whether most of the performance gain arises from modifying a single pathway. The results in Table 7 show that each isolated component provides measurable improvement over

the baseline, with Full PEFT achieving the highest classification metrics while remaining parameter-efficient.

## Appendix F. Representation Analysis

### F.1. Original VLM Embedding & Cosine Similarity Matrix

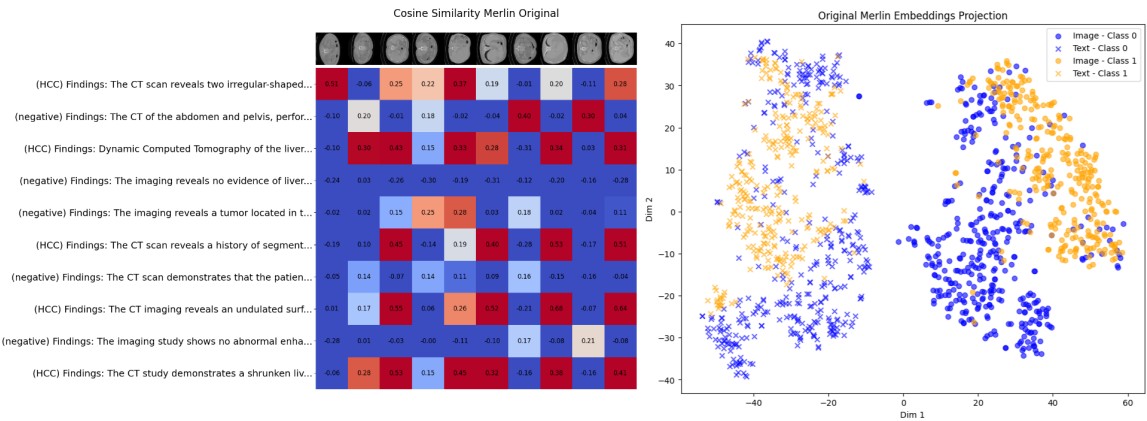

Figure 7: *Left*: Original VLM Cosine Similarity Matrix *Right*: Original VLM Embedding Distribution

Before adaptation, the pretrained VLM exhibits limited separation between HCC and non-HCC cases in both the cosine-similarity matrix and embedding space. As shown in Figure 7, similarity scores are diffuse, and class structure is weak, reflecting the model's lack of liver-specific supervision

### F.2. Cosine-Similarity Matrices

The cosine-similarity matrices in Figure 8 - 11. illustrate how contrastive temperature shapes the structure of the similarity space. At low initial temperature $\tau = 0.07$, the model produces sharper but unstable similarity distributions, leading to a weaker separation between HCC and negative samples. As the initial temperature increases, the model becomes more confident in distinguishing positive from negative cases, resulting in clearer separation between different classes. However, very high initial temperatures (e.g. $\tau = 10.0$) also cause **intra-class similarity inflation**: the model begins to treat all HCC reports as highly similar to one another, regardless of their underlying clinical variation. This reflects the model learning coarse class-level semantics of "HCC vs non-HCC" while losing the ability to differentiate among individual positive samples. Consequently, the cosine similarity values among distinct HCC reports increase, reflecting a collapse toward a single-class prototype rather than preserving finer distinctions. This phenomenon aligns with the embedding-space observations and highlights the trade-off between inter-class separability and intra-class granularity at high initial temperatures.

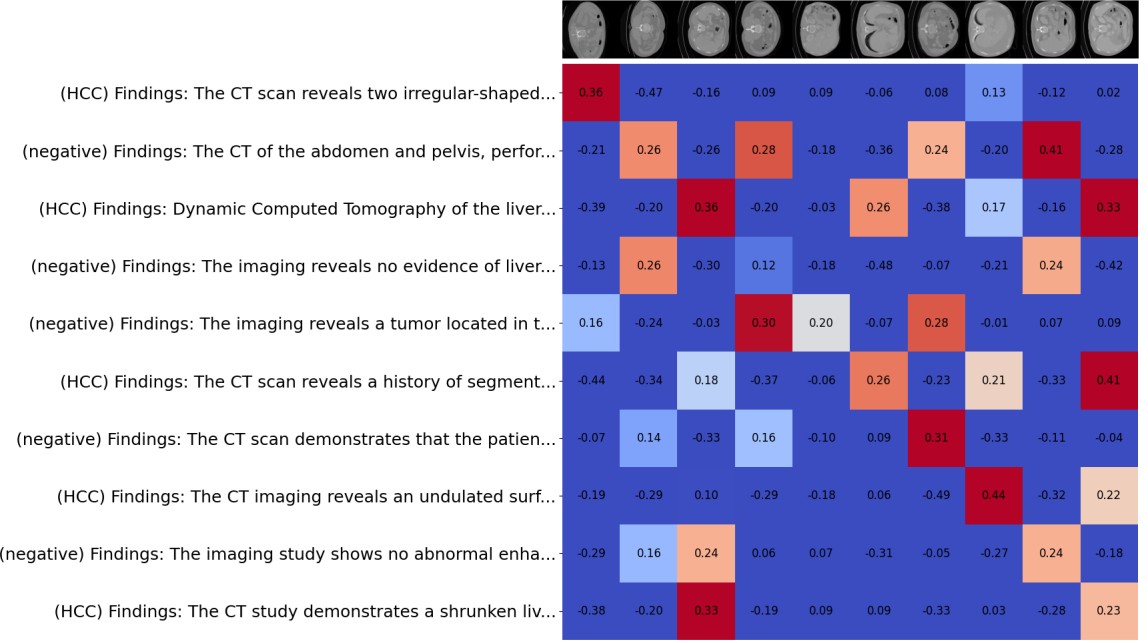

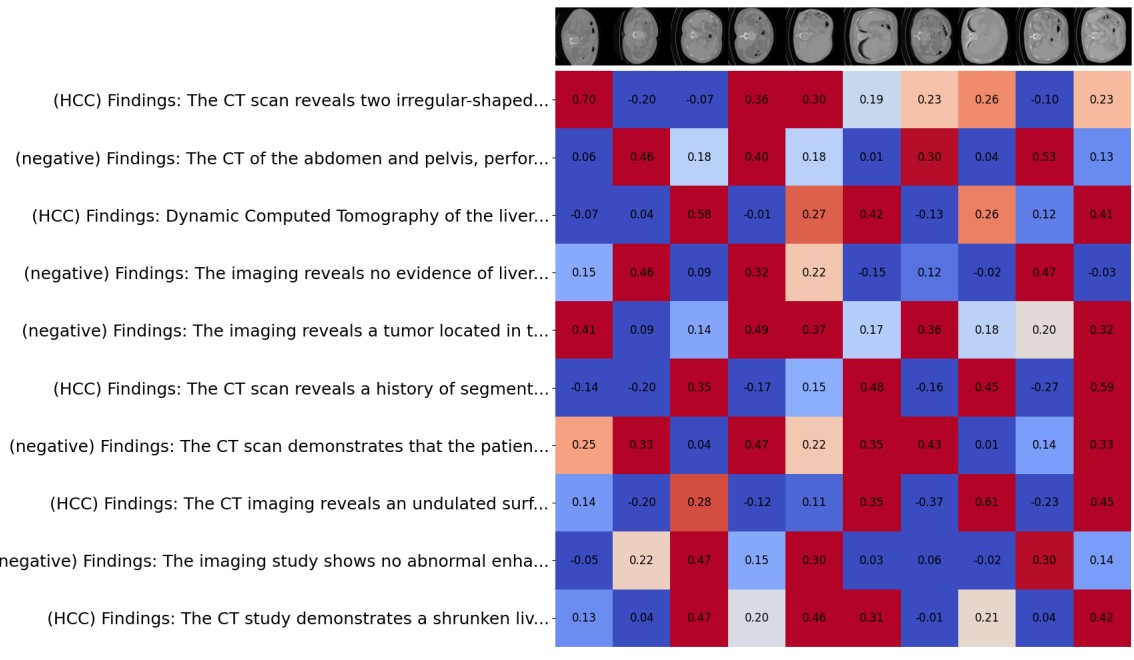

Figure 8: Cosine Similarity Matrices with initial temperatures [**0.07**].

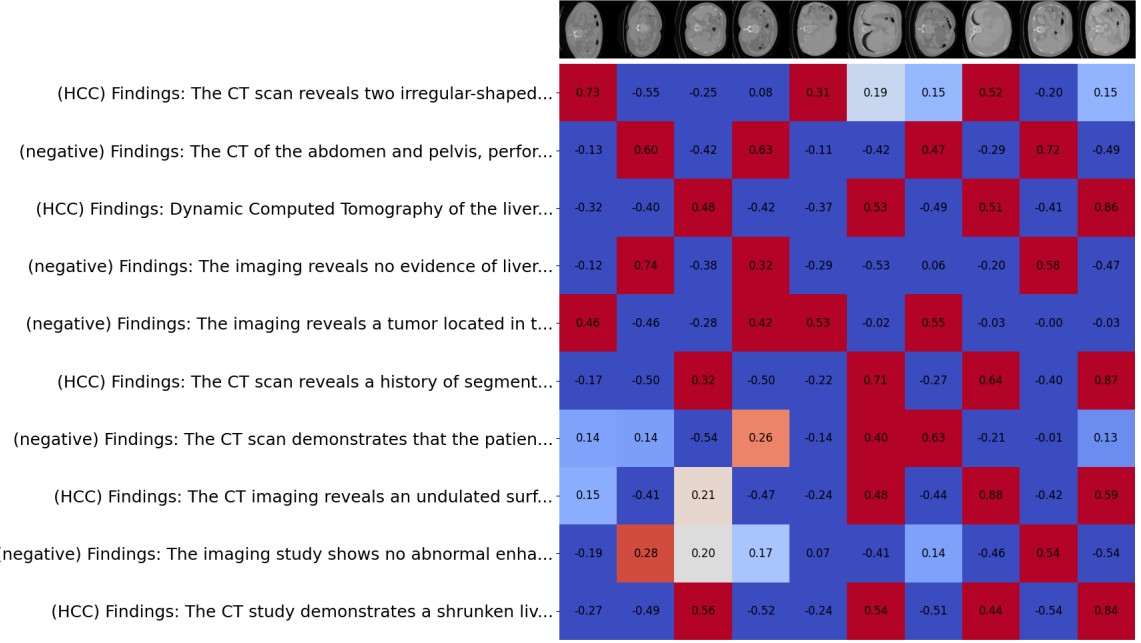

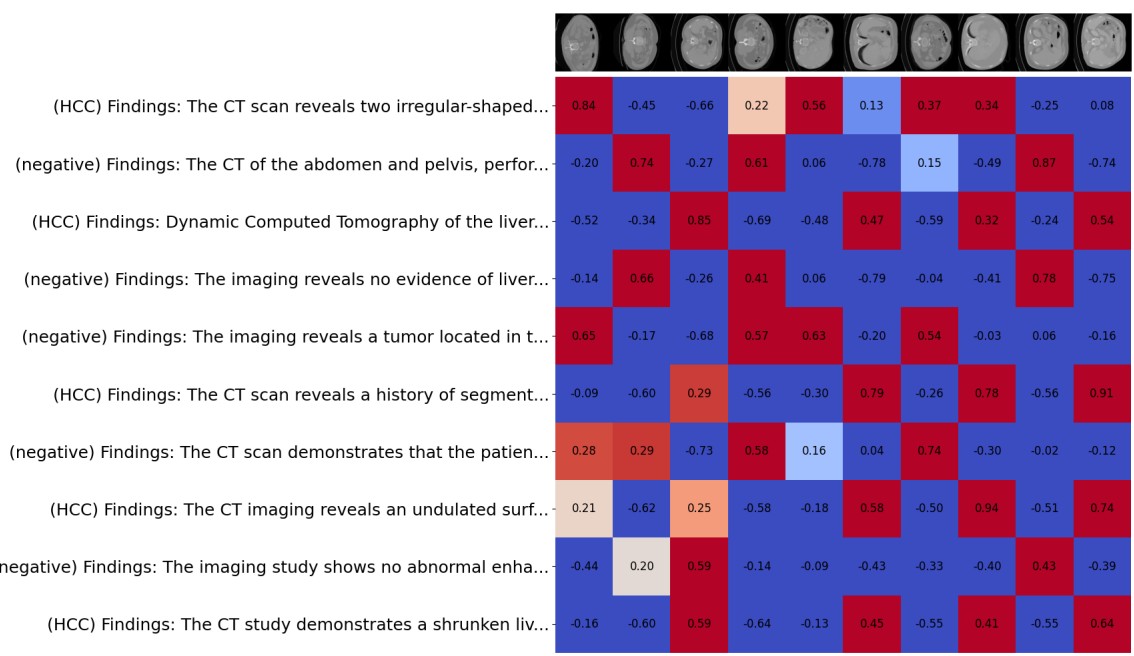

Figure 9: Cosine Similarity Matrices with initial temperatures [**0.5**].

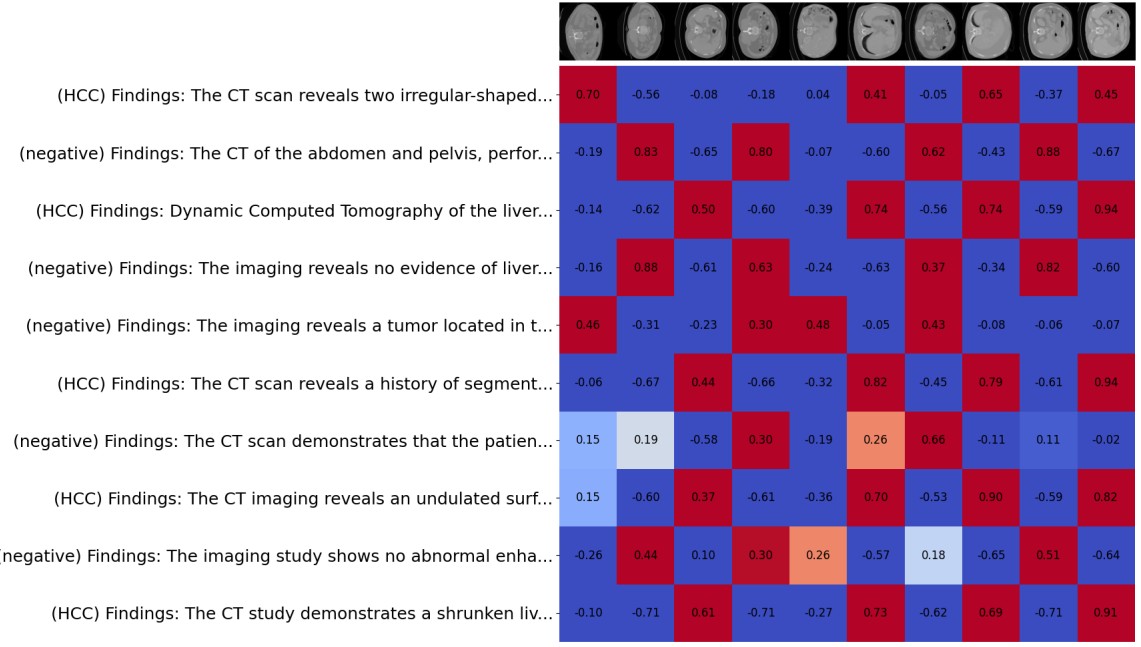

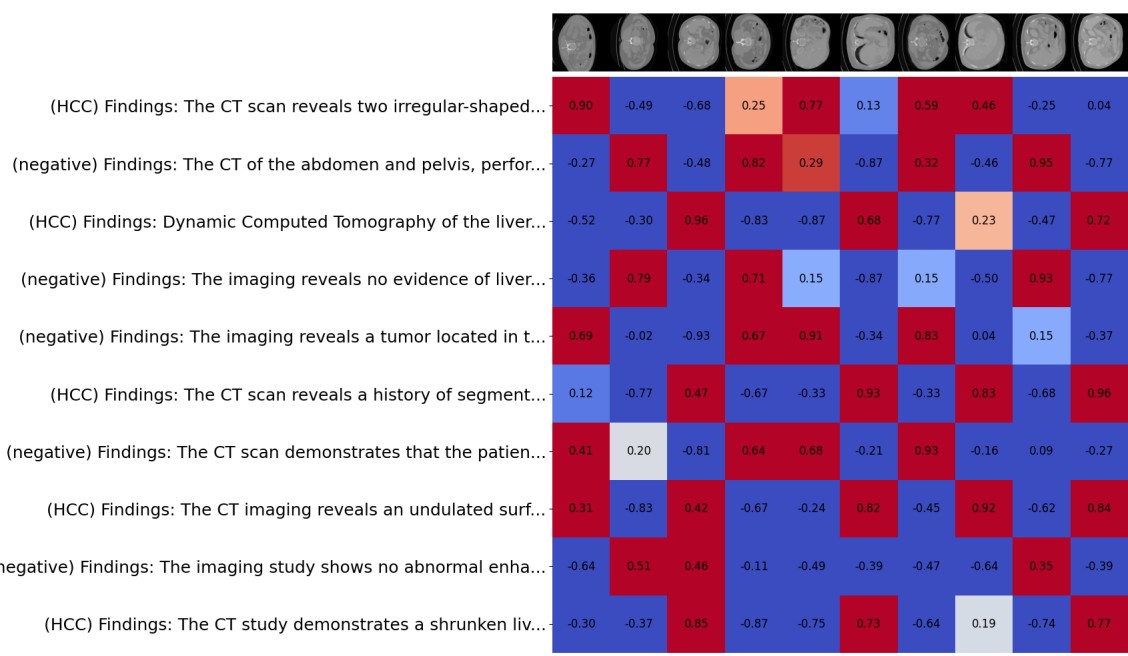

Figure 10: Cosine Similarity Matrices with initial temperatures [**1.0**].

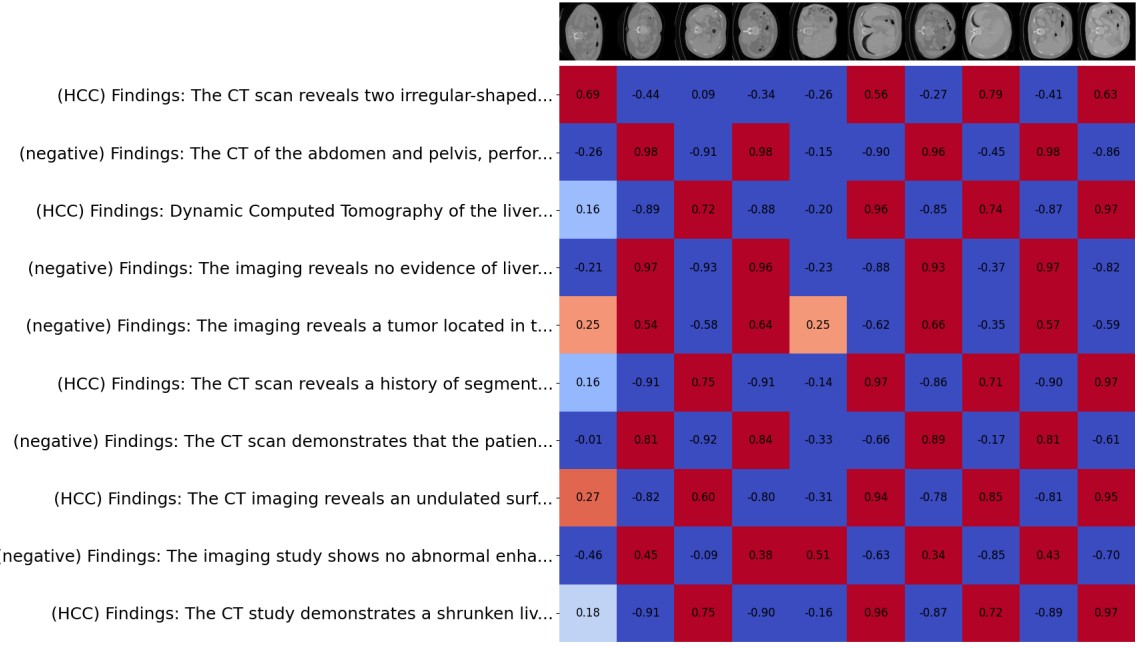

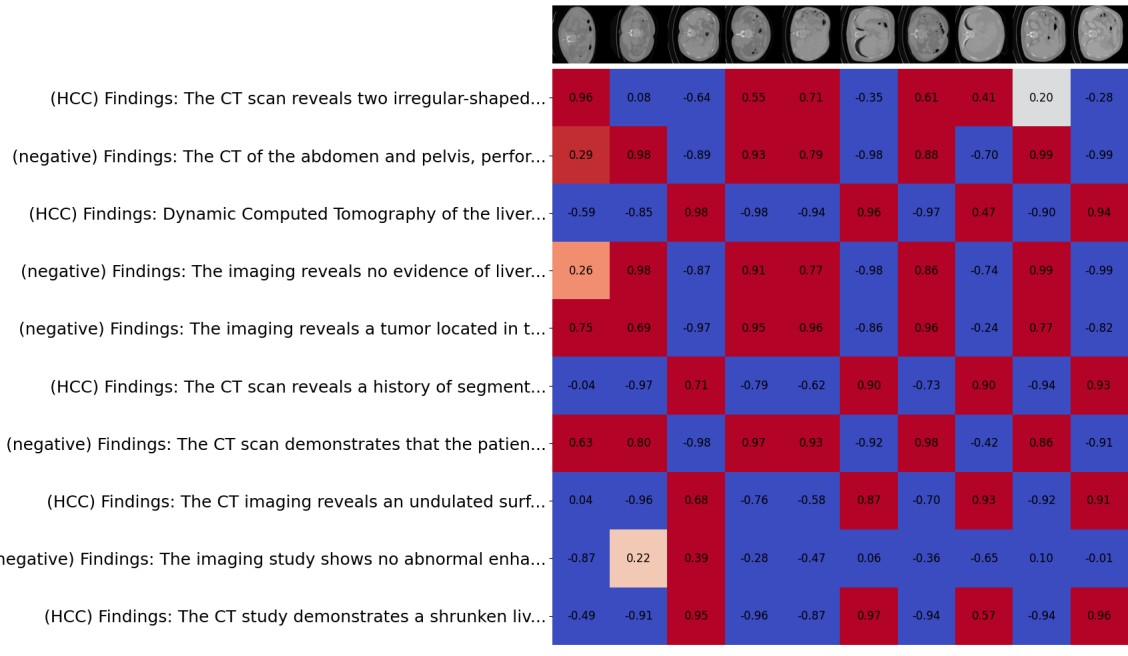

Figure 11: Cosine Similarity Matrices with initial temperatures [**10.0**].

## F.3. Softmax Temperature Distribution

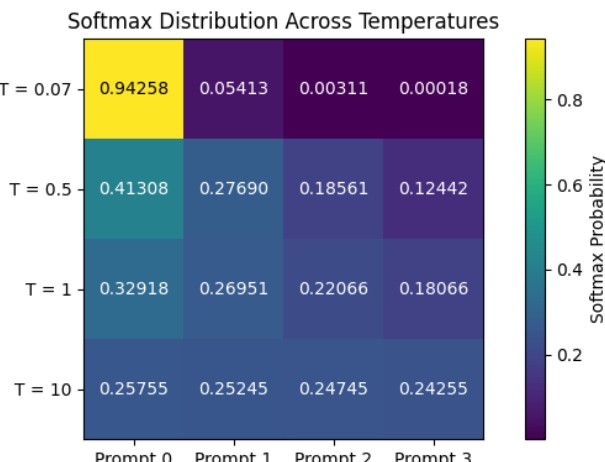

Figure 12: Illustration of different contrastive temperatures ($\tau$) initialization affect the sharpness of the softmax similarity distribution.

Although temperature setting is inherited from CLIP-style (Radford et al., 2021) contrastive learning, it remains largely unexamined in existing 3D medical VLMs. Our results show that temperature initialization directly shapes adaptation dynamics by controlling the penalties applied to hard negative samples in the InfoNCE loss (van den Oord et al., 2019). Lower temperatures initialization creates sharper similarity distributions that amplify these penalties, often leading to unstable updates and overfitting, whereas higher temperatures promote smoother gradients, more coherent embedding structure, and clearer class separation.

