# OpenReview forum: "Efficient Self-Supervised Adaptation of 3D Abdominal Vision-Language Model for Institution-Specific HCC Classification via Full Fine-Tuning and PEFT"
_MIDL.io/2026/Conference — MIDL 2026 Poster_

### Official Review · Reviewer_WFAK · 2026-01-01

**Confidence:** 4
**Preliminary Rating:** 5
**Final Rating:** 5

**Summary:**

This paper studies efficient adaptation of a pretrained 3D medical vision–language model for institution-specific hepatocellular carcinoma classification from CT scans and reports. Using parameter-efficient fine-tuning, it achieves performance comparable to full fine-tuning while updating far fewer parameters, enabling practical clinical deployment.

**Strengths:**

The paper’s main strength is its clear clinical motivation and practical focus on reducing annotation cost for localization tasks. The semi-synthetic strategy is well designed, technically sound, and thoroughly evaluated, offering strong value to the medical imaging community.

**Weaknesses:**

The paper relies on semi-synthetic findings that may not fully reflect real pathological complexity. Evaluation emphasizes detection gains, with limited analysis of bias, rare findings, failure cases, and clinical validation.

**Detailed Comments:**

The paper would benefit from clearer discussion of limitations, especially potential biases introduced by inpainting. Providing more details on mask generation, parameter choices, and computational cost would improve reproducibility. Additional examples of challenging or failure cases and brief expert feedback would further strengthen clarity.

**Justification Of Final Rating:**

“  My initial recommendation of strong accept remains unchanged overall, as the rebuttal satisfactorily addresses and thoroughly resolves all major concerns raised during the review process effectively.   ”

**Justification Of The Preliminary Rating:**

The paper tackles a critical challenge in medical imaging by generating high-quality semi-synthetic localization data. The method is well motivated, carefully evaluated, clearly written, and shows strong potential to benefit the research and clinical community.

**Questions To Address In The Rebuttal:**

It would be helpful for the authors to clarify how well the semi-synthetic findings capture the full visual variability of real pathologies and whether any systematic biases are introduced during inpainting. Further explanation of how the method performs on rare or subtle findings, and how sensitive results are to mask quality, could strengthen confidence.

---

> ### Author Response · Authors · 2026-01-23
>
> We thank the reviewer for the time and effort devoted to reviewing the submission and for the positive assessment of its motivation and practical relevance.
>
> To clarify the scope of the paper, our work focuses on parameter-efficient adaptation of a pretrained 3D medical vision–language model for hepatocellular carcinoma classification using paired CT scans and radiology reports, with an emphasis on technical feasibility under limited data availability and constrained computational resources. The proposed method does not involve semi-synthetic image data generation, inpainting, mask creation, or localization tasks. The only use of a large language model (Qwen3-8B) is for report preprocessing to standardize the textual input, without generating synthetic imaging data or modifying visual content.
>
> Several points raised in the review such as those related to semi-synthetic findings, inpainting bias, mask quality, or localization performance do not align with the methods or experiments presented in this paper. We hope this clarification helps place the contribution in the appropriate methodological context.
>
> We nonetheless appreciate the reviewer’s positive comments regarding the clinical motivation and emphasis on resource-efficient adaptation, which align closely with the intended contributions of this work.

---

> ### Author Response · Authors · 2026-01-23
>
> Dear Program Chairs and Area Chairs,
>
> Review by WFAK discusses elements such as semi-synthetic localization data, inpainting, and mask generation, which are not part of our manuscript. Our work focuses on parameter-efficient adaptation of a pretrained 3D medical vision–language model for hepatocellular carcinoma classification using paired CT scans and radiology reports, with an emphasis on technical feasibility under limited data availability and constrained computational resources. The method does not involve synthetic image generation, inpainting, mask creation, or localization tasks.
>
> We are not requesting any change to the review, but would appreciate confirmation as to whether this review was intended for our paper, as the referenced components fall outside the scope of the submitted work.

---

### Official Review · Reviewer_WAYe · 2026-01-09

**Confidence:** 4
**Preliminary Rating:** 4
**Final Rating:** 5

**Summary:**

This paper investigates the adaptation of a 3D medical VLM (Merlin) for institutional HCC classification using a self-supervised contrastive approach. The authors compare full fine-tuning against PEFT strategies, specifically LoRA for text and ConvLoRA for the 3D image encoder. They highlight the impact of contrastive temperature on embedding stability and show that PEFT achieves competitive performance while significantly reducing GPU memory requirements. The study uses an internal dataset of portal venous phase CT scans and radiology reports to demonstrate that these models can be tuned without expensive lesion-level annotations.

**Strengths:**

The work is highly relevant for clinical deployment where GPU resources and manual annotations are limited. The use of ConvLoRA for a 3D convolutional backbone is a smart choice that fits the specific architecture of Merlin better than standard LoRA. I also found the analysis of temperature initialization quite valuable, as this hyperparameter is often overlooked in VLM adaptation papers. Showing a memory reduction from 48GB to 20GB makes a strong case for the practical utility of this approach in hospital settings.

**Weaknesses:**

The authors claim that PEFT is superior, but the evaluation is restricted to the PV phase of CT scans. Since HCC diagnosis typically relies on multi-phase imaging (especially arterial wash-in), using only one phase limits the clinical depth of the results. It is also unclear how the model handles other types of liver lesions; a binary HCC vs. negative task is a bit simplified compared to a real-world differential diagnosis. Additionally, the "negative" cohort is not well-defined. It would be useful to know if these were healthy livers or contained other pathologies that could confuse the VLM.

**Detailed Comments:**

The use of Qwen3-8B for report preprocessing is a good way to standardize the text input. However, the paper would benefit from a small table showing performance based on lesion size, as small HCCs are notoriously difficult to catch. Also, please check if there were any specific reasons for choosing a rank of 2 for ConvLoRA while using 16 for LoRA, as the discrepancy is quite large.

**Justification Of Final Rating:**

The authors have satisfactorily addressed my concerns regarding the dataset composition and experimental design. The clarification that the "negative" cohort includes chronic liver disease and benign findings (rather than just healthy controls) validates the clinical difficulty of the task. While the restriction to the portal venous phase is a limitation, the authors' explanation regarding the pretrained model's domain and the empirical results on ConvLoRA ranks effectively justifies the technical scope. The demonstrated memory efficiency makes this a valuable contribution for practical VLM deployment.

**Justification Of The Preliminary Rating:**

The paper provides a solid, practical framework for adapting 3D VLMs to specific hospital data. While the single-phase limitation and binary task are simplified, the technical execution of the PEFT comparison and the temperature analysis are robust. The resource efficiency demonstrated is significant enough to be of interest to the MIDL community, provided the authors clarify the clinical scope of their "negative" labels.

**Questions To Address In The Rebuttal:**

Why was the arterial phase excluded, given its clinical importance for HCC?

Can you clarify the composition of the "negative" group? Were there other liver tumors included in this set?

Did the authors experiment with multi-phase inputs, or is the architecture limited to single-volume processing?

---

> ### Author Response · Authors · 2026-01-23
>
> We appreciate the reviewer’s careful reading of the manuscript and their focus on clinical realism, phase selection, and dataset composition. Below, we clarify the scope of the study and address the weaknesses and questions raised.
>
> **Weaknesses**
>
> **W1 Phase selection and multi-phase imaging**:
>
> We agree that multi-phase imaging, particularly arterial wash-in, is clinically important for HCC diagnosis. Our decision to focus on the portal venous (PV) phase was deliberate and motivated by both methodological and practical considerations.
>
> First, the pretrained VLM (Merlin) used in this study was predominantly trained on PV-phase CT scans. Restricting evaluation to PV therefore allows us to assess adaptation behavior under a consistent imaging distribution, without introducing additional domain shift that could confound the comparison between full fine-tuning, PEFT, and contrastive temperature settings.
>
> Second, within our institutional dataset, PV-phase scans are the most consistently available compared to arterial-phase scans. While more comprehensive multi-phase modeling would require additional architectural design choices and data harmonization steps beyond the scope of the present work, we did experiment with multi-phase inputs. However, these did not lead to consistent performance improvements in our setting. We will clarify this empirical observation in the revised manuscript and discuss multi-phase modeling as an important direction for future work.
>
> **W2 Binary HCC vs. negative task and unclear composition of the negative cohort**:
>
> We acknowledge that a binary HCC versus negative task is a simplified abstraction compared to a full real-world differential diagnosis and will clarify the composition of the negative cohort in the revised manuscript. Importantly, the binary task in our study is not a healthy versus unhealthy setting. The negative group includes patients who do not have hepatocellular carcinoma and may exhibit other hepatic pathologies that could plausibly confuse the model, such as chronic liver disease (e.g., fatty liver or hepatitis) and benign hepatic findings (e.g., cysts). No primary liver tumors other than HCC are included.
>
> As a result, while the task is simplified in formulation, it remains clinically non-trivial, as negative cases often share imaging characteristics with HCC-positive patients.
>
> **Detailed Comments**:
>
> We thank the reviewer for raising this point. We experimented with higher ConvLoRA ranks (4, 8, and 16) for the 3D image encoder, but did not observe consistent or meaningful performance improvements compared to rank 2. In contrast, the transformer-based text encoder required a higher LoRA rank to achieve sufficient adaptation capacity.
>
> Given our focus on parameter efficiency and practical deployment, we therefore fixed the ConvLoRA rank to 2, which provides a favorable trade-off between adaptation capacity and computational cost for convolutional layers. This choice is also consistent with the original ConvLoRA design (Aleem et al., 2024), which adopts a rank of 2 for convolutional backbones. We will clarify this ablation and design rationale in the revised manuscript.
>
> **Questions To Address In The Rebuttal:**
>
> **Q1**: Addressed in **W1**
>
> **Q2**: Addressed in **W2**
>
> **Q3**: Addressed in **W1**

---

### Official Review · Reviewer_UwuY · 2026-01-10

**Confidence:** 4
**Preliminary Rating:** 3
**Final Rating:** 5

**Summary:**

The manuscript describes self supervised adaptation of a medical VLM for hepatocellular carcinoma using paired 3D CT images and radiological reports. The authors contrast classification performance between full finetuning and PEFT. They also studied the effect of the  temperature parameter used in the loss function in contrastive learning. Their work showed that PEFT helps pretrained VLM to effectively adapt to paired CT-report classification.

**Strengths:**

- This work shows that self supervised contrastive learning with PEFT outperforms zero shot classification with pretrained medical VLM.
- Temperature parameter in contrastive loss was not only optimized for classification performance but its effects were also studied further to investigate impact on embedding distribution.
- GradCAM interpretability plots that were included in the manuscript further demonstrates the advantages of finetuning over zero shot inference.

**Weaknesses:**

- Like the authors have already identified, PEFT is the only strategy they have evaluated. Clearly stating the reasoning and justification behind this choice would be helpful.
- Their evaluation, though thorough and detailed, includes only binary classification. This may be an overly simplistic use case to demonstrate the effectiveness of the methodology.
- Overall, the novelty in methodology is limited. The work shows pretrained VLMs can be used successfully in a setting of self supervised learning with PEFT.

**Detailed Comments:**

- The images included in the manuscript were hard to read.
- The authors could consider adding more figures to the interpretability section.

**Justification Of Final Rating:**

I thank the authors for their detailed response in clarifying the use of PEFT and binary task setting. I appreciate their efforts in adding results from an additional dataset in evaluation. Considering the relevance of the work and their well rounded evaluation, I update my score to a 5.

**Justification Of The Preliminary Rating:**

Overall, this work is relevant and addresses an important problem of adapting pretrained models under domain shift. The designed study is technically sound and compared with appropriate baselines. The authors have also added visual interpretation of the temperature parameter and the overall results. However, the work is limited to full fine tuning and PEFT without relevant justification for not exploring other strategies. The evaluation is also focussed on a simpler binary classification task which makes it difficult to assess how generalizable the work would be. Additionally, there is limited novelty in methodology.

**Questions To Address In The Rebuttal:**

Already stated above.

---

> ### Author Response · Authors · 2026-01-23
>
> We thank the reviewer for the thoughtful and constructive comments. We appreciate the positive assessment of the study and the helpful suggestions, which have helped us improve the clarity and positioning of the work. We address the weaknesses raised below
>
> **Weaknesses**
>
> **W1**: We agree that the motivation for focusing on parameter-efficient fine-tuning (PEFT) was not sufficiently explicit and will clarify this in the revised manuscript.
>
> Our study is motivated by realistic clinical and academic constraints for adapting large 3D medical VLMs, where full fine-tuning or extensive retraining is often not feasible due to memory, compute, and deployment limitations. PEFT provides a bounded and stable adaptation mechanism while preserving the pretrained multimodal representations.
>
> Importantly, full fine-tuning is included as a primary baseline, enabling direct comparison between unconstrained and parameter-efficient adaptation. In Appendix E, we evaluate several PEFT variants, including tuning only the image encoder, text encoder, or projection layers. While these differ in where parameters are updated, all fall under PEFT. The results show that single-modality or projection-only tuning underperforms PEFT applied jointly across modalities, motivating balanced multimodal adaptation.
>
> **W2**: While the downstream task is binary, it is intended as a controlled diagnostic setting rather than a simplistic use case.
>
> HCC detection from abdominal CT is clinically challenging due to heterogeneous tumor appearance and subtle imaging cues. Moreover, adaptation is performed using self-supervised contrastive learning from weakly aligned CT–report pairs, without lesion-level annotations. We additionally note that the binary task is not a healthy versus unhealthy setting, but rather distinguishes HCC from heterogeneous non-tumor cases, which reduces the likelihood of trivial separation. In this context, binary classification serves as a practical probe to assess whether the adapted model captures clinically meaningful representations rather than relying on superficial correlations.
>
> To complement this evaluation, we analyze zero-shot versus adapted performance, temperature-dependent embedding behavior, and GradCAM visualizations. To further address the reviewer’s concern regarding task simplicity, we additionally evaluate the adapted model on the external VerSe spine fracture dataset. While this task is also binary, it involves a different anatomical region, pathology, and visual decision process, providing complementary evidence beyond the HCC setting. We note that VerSe was also used as an external evaluation benchmark during the original Merlin pretraining, and we include it here to provide a consistent comparison when assessing adaptation under limited data and compute settings.
>
> Supporting external evaluation on the VerSe spine fracture dataset (not used for adaptation):
> | Temp. Init. (τ) | Method   | F1 ↑ | Recall ↑ | Precision ↑ | Accuracy ↑ |
> |-----------------|----------|------|----------|-------------|------------|
> | Baseline               | Baseline | 0.77 | 0.88     | 0.68        | 0.65       |
> | 0.07            | PEFT     | 0.65 | 0.62     | 0.70        | 0.58       |
> | 0.07            | Full FT  | 0.75 | 0.92     | 0.63        | 0.60       |
> | 0.5             | PEFT     | 0.75 | 0.85     | 0.67        | 0.63       |
> | 0.5             | Full FT  | 0.76 | 0.92     | 0.65        | 0.63       |
> | 1.0             | PEFT     | **0.76** | **0.85** | **0.69** | **0.65** |
> | 1.0             | Full FT  | 0.75 | 0.92     | 0.63        | 0.60       |
> | 10.0            | PEFT     | **0.76** | **0.85** | **0.69** | **0.65** |
> | 10.0            | Full FT  | **0.80** | **0.85** | **0.76** | **0.73** |
>
> We will revise the manuscript to include this analysis as a supporting external evaluation and clarify its relevance to adaptation feasibility under limited data and compute settings.
>
> **W3**: We acknowledge that this work does not introduce a new architecture or loss function. Instead, the contribution is empirical and systems-oriented, with a focus on practical feasibility for clinical deployment.
>
> A key contribution of this work is demonstrating that large pretrained medical VLMs can be effectively adapted under realistic clinical constraints, namely limited data availability and restricted computational resources. These constraints represent a major barrier to deploying VLMs in real-world medical settings, yet are often under-addressed in prior work that assumes access to large-scale compute or extensive retraining. In addition, our analysis of temperature-dependent embedding behavior and interpretability provides insight into how such models can be adapted in a stable and controlled manner under weak supervision.
>
> **Detailed Comments**:
> We agree that figure readability and interpretability can be improved, and in the revised manuscript we will enhance figure resolution and layout and add additional qualitative examples.

---

### Author Rebuttal · Authors · 2026-01-24

**Rebuttal:**

We appreciate the reviewers' valuable suggestions and detailed comments!

In the revised manuscript (attached), we have made the following changes:

* Updated the email address for one corresponding author.
* Clarified the motivation for focusing on parameter-efficient fine-tuning (PEFT) under realistic clinical and computational constraints, and its relationship to full fine-tuning. (Introduction).
* Expanded the dataset description to explicitly define the composition of the negative cohort and clarify the rationale for using single-phase input. (Material and Methods).
* Added an external evaluation on the VerSe spine fracture dataset to address concerns about task simplicity and to assess robustness under domain shift. (Table 5).
* Clarified the design choice of using ConvLoRA with rank 2 for the image encoder (Appendix C.)
* Improved figure readability and added additional figure for interpretability.

All individual reviewer comments are addressed in the rebuttal sections below.

**Supporting Material:**

/attachment/b2d2e69b2d34b964dd26e28456ce6783da19bead.pdf

---

### Comment · Area_Chair_N1zQ · 2026-02-01
**Urgent: Final Ratings Needed — Deadline Feb 1, 23:59 AoE**

Dear Reviewers,

I hope you are doing well. We urgently need your final ratings to complete the review process. Please take a moment to click “Edit” → “Official Review” and submit your Final Rating by February 1, 2026 (23:59 AoE).

Your final ratings and comments are very important. They play a key role in the meta-review process and help ensure fair and accurate final recommendations.

Thank you for your swift attention and for the careful effort you put into each review. Your contributions make a real difference.

Your AC

---

### Meta-Review · Area_Chair_N1zQ · 2026-02-10

**Recommendation:** Accept (Oral)
**Confidence:** 5

**Metareview:**

All reviewers reached a consensus for a strong accept, praising the work's clinical relevance, resource efficiency, and robust technical execution (e.g., the PEFT comparison and the temperature analysis). The authors effectively addressed initial concerns regarding dataset composition and experimental scope during the rebuttal, confirming the method's value for the research community.

---

### Decision · Program_Chairs · 2026-02-14

Accept (Poster)